# Deeply Learned Spectral Total Variation Decomposition

**Tamara G. Grossmann**
DAMTP
University of Cambridge
Cambridge, UK
tg410@cam.ac.uk

**Yury Korolev**
DAMTP
University of Cambridge
Cambridge, UK
yk362@cam.ac.uk

**Guy Gilboa**
Department of Electrical Engineering
Technion - Israel Institute of Technology
Haifa, Israel
guy.gilboa@ee.technion.ac.il

**Carola-Bibiane Schönlieb**
DAMTP
University of Cambridge
Cambridge, UK
cbs31@cam.ac.uk

## Abstract

Non-linear spectral decompositions of images based on one-homogeneous functionals such as total variation have gained considerable attention in the last few years. Due to their ability to extract spectral components corresponding to objects of different size and contrast, such decompositions enable filtering, feature transfer, image fusion and other applications. However, obtaining this decomposition involves solving multiple non-smooth optimisation problems and is therefore computationally highly intensive. In this paper, we present a neural network approximation of a non-linear spectral decomposition. We report up to four orders of magnitude ($\times 10,000$) speedup in processing of mega-pixel size images, compared to classical GPU implementations. Our proposed network, TVSpecNET, is able to implicitly learn the underlying PDE and, despite being entirely data driven, inherits invariances of the model based transform. To the best of our knowledge, this is the first approach towards learning a non-linear spectral decomposition of images. Not only do we gain a staggering computational advantage, but this approach can also be seen as a step towards studying neural networks that can decompose an image into spectral components defined by a user rather than a handcrafted functional.

## 1 Introduction

Transforming and processing information such as images in a frequency domain to facilitate analysis and manipulation is a classical and very successful approach. A prominent example of a linear spectral decomposition is the Fourier transform that uses the trigonometric basis to represent a signal or an image. However, this linear transform is not optimal for images that contain discontinuities (edges), which can only be represented using high frequencies. To overcome this, a non-linear spectral decomposition based on the edge-preserving total variation (TV) functional was proposed in [17, 18]. The TV transform enables a scale representation based on the size and contrast of the structures contained in an image. The spectral components are related to eigenfunctions induced by the TV functional such as indicator functions of disks and other smooth convex shapes. Similarly to the linear case, the spectral components of an image defined by the non-linear TV transform can be filtered, extracted and attenuated at different scales. Manipulation and analysis of images using the spectral TV decomposition has found various successful applications ranging from image

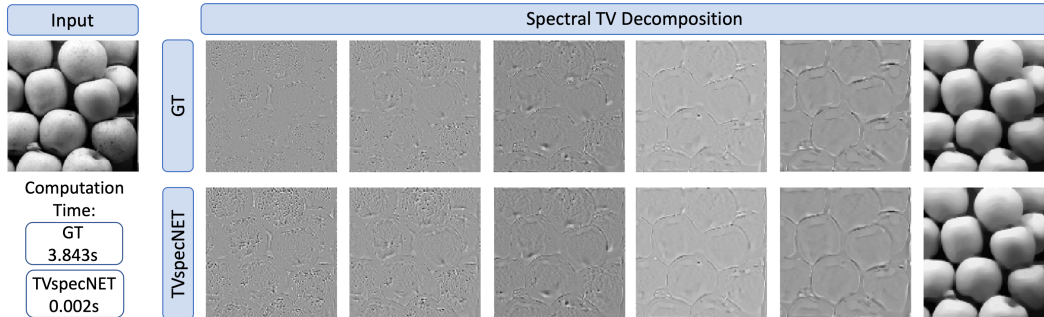

Figure 1: Visual comparison of our proposed TVspecNET decomposition and the ground truth (GT) [18] on an example image from MS COCO [32]. For this image, the resulting evaluation measures are: SSIM: 0.9849, PSNR: 32.046, sLMSE: 0.6062

denoising [37] through texture extraction and separation [6, 24] and image fusion [4, 48, 21, 34] to non-linear segmentation in biomedical imaging [45]. Especially in image fusion applications, spectral TV decomposition is able to overcome challenges in relation to edge and detail preservation where other methods fail [48]. The theory of non-linear spectral decomposition has been extended to arbitrary one-homogeneous functionals in [8, 9, 7] and $p$-homogeneous functionals ($p \in (1, 2)$) in [14]. Recently, spectral TV decomposition has additionally been generalised from the Euclidean space to surfaces [15].

In order to obtain the spectral TV decomposition of an image, the solution to the TV flow needs to be computed at every scale. This involves solving multiple non-smooth optimisation problems and is therefore computationally costly. To overcome this issue, we consider training a neural network (NN) to reproduce the spectral TV decomposition at a considerably reduced computational cost. Since the TV transform can be seen as a non-linear analogue to the Fourier transform, we aim to obtain an analogue to the Fast Fourier Transform (FFT) in our approach. The availability of a fast computational method is arguably one of the reasons for the success of the Fourier transform in signal and image processing. Fast methods for non-linear spectral decomposition therefore have the potential to become an equally important tool for image and data analysis.

Decomposing images into task dependent components via deep learning has been used in multiple imaging tasks, such as denoising [46, 33, 47] (components are the noise-free image and noise), segmentation [41, 2, 16], material decomposition [43, 36, 30] (e.g. separation to bones and soft-tissues in medical imaging) and intrinsic image decomposition [38, 26, 29, 31] (shading and reflectance). The task of obtaining a non-linear spectral decomposition is, however, different. While in the examples above, the components are defined semantically, i.e. they depend on the contents of the image, spectral TV decompositions are based on a PDE, hence to learn a spectral decomposition, the network has to implicitly learn a PDE. This allows one to apply the trained network to images significantly different from the training set.

Training a NN that reproduces an analytical spectral decomposition based on a handcrafted functional is a first step towards an even more ambitious goal of learning user defined (data driven) decompositions. This would involve training a NN to reproduce the desired behaviour on user defined 'eigenfunctions', which can be then transferred to real images.

**Contributions**  In this paper, we propose a neural network that we call TVspecNET that can reproduce the spectral TV decomposition of images while significantly (by more than three orders of magnitude) reducing the computation time to obtain the decomposition once the network is trained. Our main contributions are as follows

- We approximate a highly non-trivial function by means of deep learning to obtain the non-linear spectral decomposition where model-driven approaches have been cumbersome and computationally complex;

- We achieve a substantial computational speed up compared to the classical, model-driven approach that is based on solving a gradient flow;

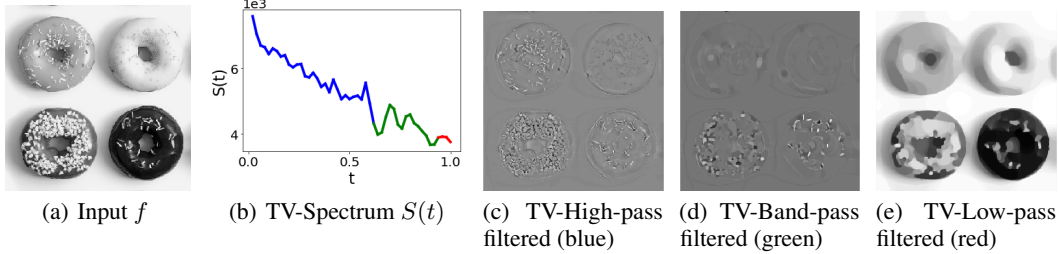

| (a) Input $f$ | (b) TV-Spectrum $S(t)$ | (c) TV-High-pass filtered (blue) | (d) TV-Band-pass filtered (green) | (e) TV-Low-pass filtered (red) |

Figure 2: Example of the filtered spectral responses of a natural image $f$ at different scales $t$. (a) the initial image with (b) the spectrum $S(t)$. TV High- (blue), band- (green) and low-pass filtered (red) spectral bands depicting small to large structures (c)–(e) separating the sprinkles from the donuts. The input image is taken from MS COCO [32].

- We demonstrate that our network is indeed capable of learning intrinsic properties of the non-linear spectral decomposition such as one-homogeneity, and rotational and translational invariance. Moreover, we demonstrate that the network not only learns the decomposition, but also implicitly learns the theoretically predicted behaviour on isolated eigenfunctions even if no isolated eigenfunctions were present in the training set. Hence the network generalises well and is able to unlock the inherent structure of the non-linear spectral decomposition;
- We perform a comprehensive comparative study that shows the optimality of our architecture for non-linear spectral decomposition;
- To the best of our knowledge, we are the first ones to propose a deep learning approach to approximate the non-linear spectral decomposition of images.

## 2 Background

### 2.1 Spectral Total Variation Decomposition

Let $\Omega \subset \mathbb{R}^N$ be a bounded image domain with Lipschitz continuous boundary $\partial\Omega$. For an initial image $f : \Omega \to \mathbb{R}$ the total variation (TV) scale-space representation can be modelled by the gradient flow of $u : [0, \infty) \times \Omega \to \mathbb{R}$:

$$u_t(t; x) = -p(t; x), \ u(0; x) = f(x), \ p(t; x) \in \partial J_{TV}(u), \tag{1}$$

where $\partial J_{TV}(u)$ denotes the subdifferential [40] at $u$ of the following convex TV functional

$$J_{TV}(u) = \sup_{\varphi} \left\{ \int_\Omega u \operatorname{div} \varphi \, dx, |\varphi|_{L^\infty} \leq 1 \right\} = \int_\Omega |Du|, \tag{2}$$

where the supremum is taken over $\varphi \in C_0^1(\Omega; \mathbb{R}^N)$ and $|\varphi|_{L^\infty} = \sup_{x \in \Omega} \sqrt{\sum_{i=1}^N \varphi_i^2(x)}$. The element $p \in \partial J_{TV}(u)$ in (1) is a subgradient of the TV functional (2) at $u$. We refer to (1) as the TV flow [1].

In order to decompose an image $f$ into its non-linear spectral components (in terms of the TV functional), [18] introduced the TV transform that is defined using the solution of the TV flow $u$ by

$$\phi(t; x) = u_{tt}(t; x)t, \tag{3}$$

where $u_{tt}$ denotes the second temporal derivative of $u$. The derivation should be understood in a weak sense (for the formal setting in spatial discrete and continuous domains see [9] and [7], respectively). The time parameter $t$ is also referred to as the *scale* and plays a role analogous to the frequency in Fourier analysis. The TV transform $\phi(t; x)$ is invariant with respect to rotations and translations of the initial image $f$. However, spatial scaling and change of contrast lead to changes in the transform domain, that is structures will be recovered in different bands [18].

For a spectral representation of an image, we generally expect the transform to generate impulses at some basic structures such as sines and cosines in the Fourier transform. In the case of total variation

these basic structures are functions $u$ satisfying $\lambda u \in \partial J_{TV}(u)$ with $\lambda \in \mathbb{R}$, which are also referred to as non-linear eigenfunctions [19]. They create an impulse at scale $t = 1/\lambda$. Examples of such eigenfunctions for the TV functional are multiples of the indicator function of a disk. The scale at which an impulse is generated for the disk depends on the radius and the height of the disk. The spectrum of the TV transform, for $\phi \in L^1$, is defined by

$$S(t) = \|\phi(t;x)\|_{L^1(\Omega)} = \int_{\Omega} |\phi(t;x)| dx,$$

and represents the $L^1$ amplitude of the spectral responses $\phi(t;x)$ at different scales $t$. An alternative definition which admits a Parseval-type equality is proposed in [9], for simplicity we will use the above definition. An example of the spectrum as well as the TV transform and the corresponding structures at different scales are shown in Figure 2.

Given the spectral responses $\phi(t;x)$, the initial image $f$ of the TV flow can be recovered through the inverse transform defined by

$$f(x) = \int_0^\infty \phi(t;x)\, dt + \bar{f}, \tag{4}$$

where $\bar{f}$ is the mean value of $f$. If we truncate the integral at some $T \in (0,\infty)$, we get

$$f(x) = \int_0^T \phi(t;x)\, dt + f_r(T;x), \tag{5}$$

where $f_r(T;x) = u(T;x) - u_t(T;x)T$ is referred to as the residual [18].

For small scales $t$ the TV transform $\phi(t;x)$ consists of structures with small spatial size and low contrast in the initial image $f$. Coarser spatial features and those with higher contrast are contained in the spectral components $\phi(t;x)$ with larger scales $t$ (cf. Figure 2). This connection between scale and features can be used to manipulate features based on scale using a filter function $H : [0, \infty] \to \mathbb{R}$

$$\phi_H(t;x) = \phi(t;x)H(t). \tag{6}$$

Substituting the filtered TV transform (6) into the inverse TV transform (4) or (5) recovers the filtered image in the spatial domain. TV-band-pass filtering for scales in $[t_{k-1}, t_k]$, $k = 1, \ldots, K$, with $t_{k-1} < t_k$, in a finite time setting can be done using

$$b^k = \int_{t_{k-1}}^{t_k} \phi(t;x)dt, \quad k = 1, \ldots, K-1; \quad b^K = \int_{t_{K-1}}^{t_K} \phi(t;x)dt + f_r(t_K, x). \tag{7}$$

The resulting filtered images $\{b^1, \ldots, b^K\}$ are referred to as bands and the initial image $f$ is then said to be decomposed into $K$ spectral bands, where for $t_0 = 0$, using (5), the following identity holds: $f = \sum_{k=1}^K b^k$.

## 2.2 Numerical Solution of the TV flow

One approach to numerically compute a spectral TV decomposition of an image $f$ is to discretise the TV flow (1). This is challenging due to the strong non-linearity and singularities in the subgradient $p$. A classical approach in the literature uses a regularised version of $p$ which reads $\mathrm{div}\left(Du/\sqrt{|Du|^2 + \varepsilon}\right)$, and uses finite differences and appropriate time-stepping for numerical approximation [13]. Here, explicit Euler time-stepping is computationally unstable, demanding a prohibitively small timestep size. More recent approaches approximate the TV flow by an implicit Euler scheme, which, as it turns out, can be realised via the following optimisation problem [18, 9]

$$u(t + dt) = \mathrm{argmin}_v \, \frac{1}{2}\|u(t) - v\|_{L^2}^2 + dt\, J_{TV}(v), \tag{8}$$

where $u(t)$ is the solution of the gradient flow at time $t$ and $dt > 0$ is the (sufficiently small) step size. The price to pay for numerical stability of the implicit Euler scheme is the non-smoothness of the optimisation problem (8), which makes it computationally expensive, even with state-of-the-art SOTA convex optimisation techniques such as PDHG or ADMM [12].

Once a solution of the TV flow (1) has been obtained, the TV transform (3) can also be discretised by finite differences and the $K$ spectral TV bands of the initial image can then be recovered by filtering. We note that such an accurate solution of the TV flow (and hence of spectral TV) requires solving (8) $N$ times, for computing $N$ time steps of size $dt$. This strongly motivates the use of alternative faster approximations. We use decompositions obtained in this manner as the ground truth of our training.

# 3 Deep Learning Approach to Spectral Decomposition

While the decomposition of an image into its TV-spectral bands gives qualitatively highly desirable results, its computational realisation is cumbersome as it amounts to the solution of a series of non-smooth optimisation problems (8). For this reason, we propose a neural network approach for obtaining a spectral image decomposition. We show that our proposed TVspecNET can approximate the spectral TV decomposition and can be computed several orders of magnitude faster than the model driven approach in Section 2.2, cf. also Figure 1.

There are several ways in which NNs can achieve a speed up in the spectral decomposition pipeline. Solving the TV flow is computationally the most expensive part. A great deal of work has recently been done on training NNs to solve PDEs, e.g. [5, 42, 3, 35, 39, 25]. While they are able to achieve good results, they require the calculation of the PDE explicitly in the loss functional. This, however, would involve an explicit expression of the subgradient $p$ of TV in (1), which is not desirable (cf. Section 2.2). Therefore, we choose to replace the whole pipeline by a NN and learn the decomposed images directly from the initial image. In that, the network has to learn the PDE only implicitly. Our problem is formulated as follows.

Given a training set of images $\{f_i\}_{i=1}^N$ and their spectral bands $\{b_i^1, \ldots, b_i^K\}_{i=1}^N$ as defined in (7), we seek to find a neural network $\Psi(\cdot, \Theta)$ with learnable parameters $\Theta$ such that

$$\Psi(f_i, \Theta) \approx \{b_i^1, \ldots, b_i^K\}.$$

A particular challenge of this problem setting is that we expect our NN to give a higher dimensional output, that is multiple spectral bands, from a single image input.

Convolutional neural networks (CNNs) have proven to be a powerful tool in many image analysis applications, e.g. [41, 46, 23]. This is due to their ability to leverage local neighbourhood information of pixels through convolutions. Some encoder-decoder type architectures such as the U-Net [41] that was originally designed for image segmentation, have been successfully applied in other decomposition tasks such as intrinsic image decomposition [26, 31, 29, 44]. However, since the TV transform depends on the interplay between spatial scale and contrast, encoder-decoder type architectures may not be well suited for spectral TV. These and other architectures that use down-/upsampling transform the spatial scale without changing contrast and hence hinder their interplay. Therefore, we use the denoising convolutional neural network (DnCNN) [46], which does not rely on down- and upsampling, as the basis of our network architecture. The DnCNN is a non-contracting feed-forward neural network that leverages network depth (i.e. multiple sequential layers) and batch normalisation together with residual learning to separate the noisy part of an image from the clean image. Our TVspecNET consists of $L = 17$ sequential convolutional layers $h_l$, $l = 1, \ldots, L$ with a rectified linear unit (ReLU) [28] activation, i.e. $\sigma(a) = \max(0, a)$. The layer can be defined by

$$h_{l+1} = \sigma(w_{l+1} * h_l + b_{l+1}), \quad l = 1, \ldots, L-1,$$

where $\Theta_L = \{w_l, b_l\}_{l=1}^L$ are the trained convolution kernels of size $3 \times 3$ and the biases. For $l = 1, \ldots, L-1$, each layer consists of 64 channels. The number of output channels in the last layer corresponds to the number of decomposed bands. This deep network design has a receptive field of $35 \times 35$ pixels and is therefore able to recover larger features.

Let $\{b^1(\Theta), \ldots, b^K(\Theta)\}$ be the output of the network $\Psi(f, \Theta)$ for input image $f$, and $\{\hat{b}^1, \ldots, \hat{b}^K\}$ the ground truth bands. To train our network, we use the normalised mean squared error (MSE) loss:

$$\mathcal{L}(\Theta) = \frac{1}{K} \sum_{j=1}^K \frac{\|b^j(\Theta) - \hat{b}^j\|_2^2}{\|\hat{b}^j\|_2^2}. \tag{9}$$

It is essential to use normalisation across the different bands in (9) to ensure that all bands contribute equally and bands with larger intensity ranges do not dominate the loss functional.

# 4 Experiments

In this section, we describe the experiments we conducted to evaluate the performance of our proposed TVspecNET. We describe the experimental setup and the evaluation scheme and demonstrate the

Table 1: Evaluation of the proposed TVspecNET on a testing dataset [32] of 1000 images against the model driven approach [18] (cf. Section 2.2). Values correspond to averages over the dataset.

|      | Average | Band 1 | Band 2 | Band 3 | Band 4 | Band 5 | residual Band |
|------|---------|--------|--------|--------|--------|--------|---------------|
| SSIM | **0.9600** | 0.9972 | 0.9906 | 0.9736 | 0.9441 | 0.8775 | 0.9771 |
| PSNR | **30.867** | 28.37 | 28.83 | 29.19 | 29.73 | 29.91 | 39.17 |
| sLMSE | **0.829** | 0.797 | 0.812 | 0.811 | 0.799 | 0.759 | 0.998 |

performance of the network. We also perform a comprehensive architecture comparison to analyse the performance that a down-/upsampling based architecture could achieve. Additionally, we provide an ablation study in the Supplemental Material to investigate different loss functionals.

## 4.1 Dataset and Training Settings

For training and testing our neural network we use the MS COCO dataset [32] that contains a large number of natural images. We take 2000 images for training and 1000 for testing. Each image is turned to greyscale and randomly cropped to a $64 \times 64$ pixel window. For the purpose of data augmentation, we also take $128 \times 128$ crops for some images and downsample them by a factor of 2, obtaining again images of size $64 \times 64$. As spectral TV decomposition is not invariant to cropping and resizing, this augmentation needs to be done during the data generation process and cannot be automated during training. After standardising the dataset to have zero mean and a standard deviation of 1, we generate $K = 50$ ground truth bands (7) using the model driven approach in Section 2.2. The bands are then combined dyadically to form 6 spectral bands. In this way, we make sure that smaller structures are decomposed in great detail while larger structures are grouped together in higher bands.

We train our network only for the first 5 bands, since the 6$^{th}$ band contains the residual $f_r$ as described in (7) and can be recovered by subtracting the sum over bands 1-5 from the initial image. We use the Adam optimiser [27] with an initial learning rate of $10^{-3}$ and multi step learning rate decay. Our neural network is trained with a batch size of 8 and for 5000 epochs on an NVIDIA Quadro P6000 GPU with 24 GB RAM.

## 4.2 Evaluation Protocol

We evaluate the performance of the network in three ways. Firstly, we give a quantitative evaluation on a testing set of 1000 natural images from the MS COCO dataset in terms of three common image quality measures, the structural similarity index (SSIM), the peak signal-to-noise-ratio (PSNR) and the inverted localised mean squared error (sLMSE) [20]. While the first two metrics are commonly used in image analysis, the sLMSE is more often found in the evaluation of intrinsic image decomposition and derives the local MSE on patches. Both for sLMSE and SSIM, the similarity between two images is high if the value is close to 1.

Secondly, we investigate whether the network is able to learn the underlying properties of the non-linear spectral decomposition that are predicted by the theory, such as one-homogeneity, and translational and rotational invariances. One-homogeneity implies that changing the contrast by a constant factor should shift the spectrum. For instance, if we multiply images by 2, we expect all image structures to appear in subsequent bands (since we use dyadic bands). Translational and rotational invariances imply that the spectral bands of a translated/rotated image experience the same translation/rotation as the original image.

To further investigate how well the network can 'understand' the TV transform, we test whether a network trained on natural images can learn eigenfunctions of the TV transform, i.e. whether it demonstrates the predicted behaviour on isolated eigenfunctions (which are very different from the images in the training set). For this purpose, we test the performance of the network on disk images.

## 4.3 Results

Firstly, we evaluate the performance of TVspecNET against ground truth decompositions obtain by solving a gradient flow (cf. Section 2.2) following the protocol in Section 4.2. A visual comparison

Table 2: Properties of the TV transform evaluated for the TVspecNET on a testing dataset of 1000 images. The comparison is made between the TVspecNET output and the expected decomposition based on one-homogeneity, and translational and rotational invariance, respectively. The values correspond to averages over the dataset.

|  | one-homogeneity | translation invariance | rotation invariance |
|---|---|---|---|
| SSIM | 0.9867 | 0.9930 | 0.9807 |
| PSNR | 33.783 | 37.593 | 32.042 |
| sLMSE | 0.880 | 0.983 | 0.885 |

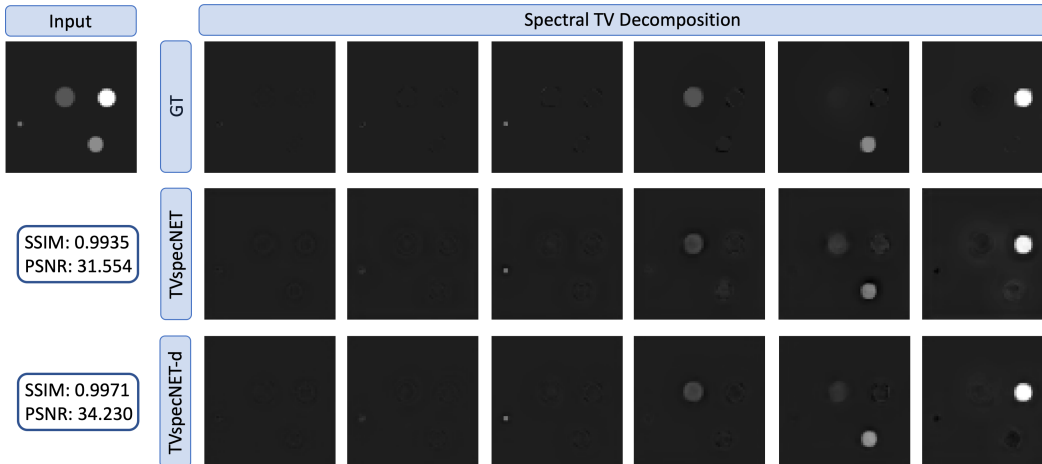

Figure 3: Visual comparison of a ground truth (GT) decomposition for a disk image and the output of TVspecNET trained only on natural images and TVspecNET-d that was additionally trained on disk images. Adding disks to the training set improves performance only slightly; even without seeing disks in the training set the network is able to learn the correct behaviour.

for an example image is shown in Figure 1. Our network is able to recover all spectral TV bands almost perfectly. This is confirmed by quantitative measures of similarity (SSIM, PSNR and sLMSE) between TVspecNET decompositions and the ground truth, as shown in Table 1.

Secondly, we demonstrate that the trained network retains properties of the TV transform predicted by the theory: one-homogeneity and translational and rotational invariances. To test translational and rotational invariances, we apply rotations/translations to the original image and then apply the network or we apply the network and then translate/rotate the bands. If the results are the same, the network has the desired invariances. To test one-homogeneity, we multiply the initial image by a factor of 2 and compare each band of the scaled image with the previous band of the original image. Since we use dyadic bands, these bands should be the same (up to the multiplication factor of 2). Quantitative results in Table 2 demonstrate an almost perfect match in all three tests. While translational invariance is inherent to fully convolutional NNs and rotational invariance is to some extent enforced through data augmentation, one-homogeneity is neither explicitly enforced nor is the network penalised to retain this property.

Thirdly, we evaluate the performance of the network on images of isolated eigenfunctions. For this test, we create images of disks with various radii and contrasts. To see how well the network learns eigenfunctions from a dataset of natural images, we make a comparison with the same architecture trained on a dataset containing both natural images and images of isolated eigenfunctions (disks). For ease of distinction, we call the second network TVspecNET-d. The results are shown in Figure 3. Remarkably, TVspecNET is capable of recovering spectral bands of disks (i.e. separating different disks into different bands) with high accuracy even without having seen them in the training set; adding eigenfunctions to the training set improves the performance only slightly. Similar observations can be made on images of ellipses. We provide further visual and quantitative examples in the Supplemental Material. Interestingly, the performance of TVspecNET-d on natural images is slightly worse compared to TVspecNET (TVspecNET-d has SSIM 0.9533, PSNR 30.622 and sLMSE 0.784).

Table 3: Computation time (in seconds) of the model driven approach evaluated on a CPU (Matlab) and on a GPU (C++/Python), and of TVspecNET with three different basis networks evaluated on a GPU. Values correspond to averages over the dataset.

| | Total number of pixels per image | | | | |
| --- | --- | --- | --- | --- | --- |
| | 4096 | 16384 | 65536 | 262144 | 1048576 |
| Model Driven on CPU | 1.6341 | 3.8431 | 12.4277 | 74.3933 | 344.6169 |
| Model Driven on GPU | 5.2104 | 5.2787 | 5.9412 | 7.8306 | 28.2811 |
| TVspecNET | **0.0020** | **0.0021** | **0.0021** | **0.0020** | **0.0020** |
| F-TVspecNET | **0.0020** | **0.0021** | 0.0022 | 0.0023 | 0.0024 |
| U-TVspecNET | 0.0057 | 0.0055 | 0.0057 | 0.0059 | 0.0062 |

Table 4: Comparison of different network architectures as the basis for our TVspecNET: DnCNN (TVspecNET), FFDnet (F-TVspecNET) and U-Net (U-TVspecNET).

| | TVspecNET | F-TVspecNET | U-TVspecNET |
| --- | --- | --- | --- |
| SSIM | **0.9600** | 0.9377 | 0.9233 |
| PSNR | **30.867** | 28.098 | 28.993 |
| sLMSE | **0.829** | 0.6854 | 0.7382 |

Finally, we compare the computation time of TVspecNET and the model driven approach in Section 2.2 and show the results in Table 3. For the model driven approach we use two implementations, a Matlab implementation [18] of the projection algorithm by Chambolle [10] running on a CPU and a primal-dual implementation [11, 22] in C++/Python[1] running on a GPU. The NN is evaluated on the same NVIDIA Quadro P6000 GPU with 24 GB RAM. As expected, the GPU implementation of the model driven approach is slower than the CPU implementation on small images (due to the GPU overhead), but becomes significantly faster for large images. TVspecNET, however, is orders of magnitude faster than both implementations on all image sizes. For the largest image size we tested, $1024 \times 1024$ pixels, the speed up of TVspecNET compared to the GPU implementation of the model driven approach is four orders of magnitude. Also, the computation time for the model driven approach increases significantly with the number of pixels (due to the increased size of the optimisation problems) while for TVspecNET it remains approximately constant.

### 4.4 Comparison of Basis Architectures

We compare different architectures as the basis of out NN: DnCNN [46] as proposed in TVspecNET, FFDnet [47] (we call this network F-TVspecNET) and U-Net [41] (we call this network U-TVspecNET). While DnCNN does not rescale or downsample the image, both FFDnet and U-Net contain pooling to various extends. FFDnet downsamples the input to 4 low resolution images before applying the network and combines the results to form a high resolution denoised image. U-Net uses multiple maxpooling and upsampling steps within the network architecture.

The results of this comparison are shown in Table 4. The proposed TVspecNET clearly outperforms both F-TVspecNET and U-TVspecNET, confirming our choice of the basis architecture. Although downsampling used in FFDnet and U-Net increases the receptive field, which is useful to recover larger features, for non-linear spectral decompositions (TV) downsampling turns into a disadvantage, since it hinders the interplay between size and contrast, which is crucial. In terms of computational time (Table 3), TVspecNET and F-TVspecNET are comparable while U-TVspecNET is approximately three times slower (still, all three are orders of magnitude faster than the model driven method).

## 5 Conclusion

In this paper, we propose TVspecNET, a neural network that can learn a non-linear spectral decomposition. We show, both through qualitative and quantitative analysis, that TVspecNET is able to decompose images into spectral TV bands. The most striking result is that, even though the

network is trained only on natural images, it is able to learn basic structures of the non-linear TV transform (eigenfunctions) and its properties such as one-homogeneity and rotational/translational invariance. Without incorporating any explicit knowledge about the underlying PDE of the spectral decomposition (TV flow) into training, this data-driven learning approach is able to learn the PDE implicitly. The speed-up that the network achieves is also impressive, going up to four orders of magnitude on $1024 \times 1024$ px images compared the state-of-the-art GPU implementation of the model driven approach. The code for TVspecNET is publicly available on Github[2]. An interesting direction for future work is 'inverting' the process and learning the decomposition from user-defined eigenfunctions, and applying the trained network to real images.

## Broader Impacts

This statement does not apply to this paper. We are concerned with computing a decomposition modelled by a PDE that can already be solved by model-driven methods, however we considerably reduce the computational cost in our approach. Therefore, we do not expect it to have immediate broader impact on society.

## Acknowledgments and Disclosure of Funding

We thank Angelica I. Aviles-Rivero, Christian Etmann, Damian Kaliroff, Lihao Liu, Tomer Michaeli and Tamar Rott Shaham for helpful discussions and advice.

This work was supported by the European Union's Horizon 2020 research and innovation programme under the Marie Skłodowska-Curie grant agreement No. 777826 (NoMADS). TG, YK and CBS acknowledge the support of the Cantab Capital Institute for the Mathematics of Information. TG additionally acknowledges support from the EPSRC National Productivity and Investment Fund grant Nr. EP/S515334/1 reference 2089694. YK acknowledges the support of the Royal Society (Newton International Fellowship NF170045 Quantifying Uncertainty in Model-Based Data Inference Using Partial Order) and the National Physical Laboratory. GG acknowledges support by the Israel Science Foundation (Grant No. 534/19) and by the Ollendorff Minerva Center. CBS acknowledges support from the Leverhulme Trust project on Breaking the non-convexity barrier, the Philip Leverhulme Prize, the EPSRC grants EP/S026045/1 and EP/T003553/1, the EPSRC Centre Nr. EP/N014588/1, European Union Horizon 2020 research and innovation programmes under the Marie Skłodowska-Curie grant agreement No. 777826 NoMADS and No. 691070 CHiPS, and the Alan Turing Institute.

We also acknowledge the support of NVIDIA Corporation with the donation of two Quadro P6000, a Tesla K40c and a Titan Xp GPU used for this research.

## Footnotes

[1]Code used from `https://github.com/VLOGroup/primal-dual-toolbox`

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
