[Supplementary Material]

# SUPPLEMENTAL MATERIAL FOR:
# Deeply Learned Spectral Total Variation Decomposition

**Tamara G. Grossmann**
DAMTP
University of Cambridge
Cambridge, UK
tg410@cam.ac.uk

**Yury Korolev**
DAMTP
University of Cambridge
Cambridge, UK
yk362@cam.ac.uk

**Guy Gilboa**
Department of Electrical Engineering
Technion - Israel Institute of Technology
Haifa, Israel
guy.gilboa@ee.technion.ac.il

**Carola-Bibiane Schönlieb**
DAMTP
University of Cambridge
Cambridge, UK
cbs31@cam.ac.uk

## A    Outline

In this document, we expand the results presented in the main paper primarily focussing on additional comparisons and new types of input images. In that, we show the optimality of our network in terms of loss functionals and highlight the generalisibility further. The document is structured as follows: First, we show visual result on disk, ellipse and natural images in Section B as well as for one-homogeneity, translational and rotational invariance. In Section C we perform an ablation study to compare the impact of different loss functionals on the results. Lastly, we give a formal definition of the quantitative metrics used for evaluation of our proposed TVspecNET in Section D.

## B    Additional Experimental Results

In addition to the results presented in the main paper, we give multiple visual examples of the TVspecNET decomposition on natural images compared to the ground truth in Figure 1 and on a disk image in Figure 2. Moreover, we consider visual and quantitative results for images of ellipses. These images neither contain isolated eigenfunctions as in disk images that we considered earlier (although isolated ellipses are eigenfunctions of the TV transform), nor are they similar to the natural images; therefore these images are somewhere in between the two types of images considered in the main paper. We generated a small dataset of 20 images that contain multiple ellipses of different size and contrast using the Python toolbox ODL (Operator Discretization Library)[1]. On this dataset, TVspecNET achieves high quantitative performance measures: SSIM is 0.9658, PSNR is 30.609 and sLMSE is 0.728, which is similar to the performance on natural images (cf. Table 1 in the main paper). We show the visual results for an example ellipse image in Figure 3.

Furthermore, we visually demonstrate one-homogeneity of TVspecNET for all image types we have so far presented. We multiply an input image by a factor of 2 and expect the resulting bands to be the same as in the original input image up to a shift into the next dyadic band (and an increase of contrast by a factor of 2). For an example image containing disks, we show the comparison between the spectral TV decomposition of the two input images obtained with TVspecNET in Figure 4(a).

(a) Decomposition of an image of a leopard. Different bands are scaled separately to better show the structures in the first bands. Image taken from the BSDS500 dataset [4].

(b) Decomposition of an image of four donuts. All bands have the same scaling. Image taken from the MS COCO dataset [3].

(c) Decomposition of an image of a giraffe. All bands have the same scaling. We show the difference between GT and TVspecNET for each band in the third row. Image taken from the MS COCO dataset [3].

Figure 1: Visual comparison of the TVspecNET decomposition and the ground truth (GT) [1] on three natural images. In (a) we scale the bands individually to better show the structures in the first bands, whilst in (b) and (c) all bands have the same scaling. In (c) we additionally show the difference between GT and TVspecNET bands.

Figure 2: Visual comparison of the TVspecNET decomposition and the ground truth (GT) [1] on an image containing multiple disks.

Figure 3: Visual comparison of the TVspecNET decomposition and the ground truth (GT) [1] on an ellipse image (first two rows). The last row displays the TVspecNET results decomposition of the ellipse image multiplied by a factor of 2. The bands get shifted as expected from a TV spectral decomposition (corresponding bands are marked by lines). The contrast was normalised, i.e. the bands in the third row were divided by 2 to match the contrast of the first two rows. This example demonstrates the one-homogeneity property of TVspecNET. The first band is split into two bands when the input is multiplied by 2. Since the last band depicts the residual, the fifth and sixth bands from the original input are combined in the residual of the decomposition of the modified image.

The one-homogeneity property for an ellipse image is demonstrated in Figure 3 in the two bottom rows. Finally, for a natural image we show the same property in Figure 4(b). Overall, we observe that one-homogeneity holds true for all image types considered in this work.

Lastly, we give example results for translational and rotational invariances of TVspecNET on a natural image in Figure 5. Spectral bands of a rotated/translated image are expected to be the same as in the original image up to the same rotation/translation.

As an extension, we trained the same neural network for a finer graded decomposition with 25 bands and the results are similarly convincing (SSIM: 0.958, PSNR: 30.228) as for dyadic bands. However, it is not necessary to retrain the network for a larger number of bands in order to obtain a finer graded decomposition of certain scales: due to one-homogeneity one can shift bands in an image by multiplying it by a constant factor (factor <1 will shift larger structures to smaller bands, a factor >1 will do the reverse). Hence, we can represent the structures of interest at a finer scale through shifting without having to retrain the network.

(a) Example on a randomly generated disk image.

(b) Example on natural image from the MS COCO dataset [3]. The contrast was normalised in this example, i.e. the bands in the second row were divided by 2 to match the contrast of the first row.

Figure 4: Visual demonstration of the one-homogeneity property on (a) a disk image and (b) a natural image. In each case, the lower row displays TVspecNET decomposition of the image in the upper row multiplied by a factor of 2. Corresponding bands are marked by lines. The first band of the original image is split into two bands when the image is scaled by a factor of 2. Since the last band contains the residual, the fifth and sixth bands from the decomposition of the original image will be combined in the residual of the scaled input.

## C   Ablation Study

In the proposed TVspecNET, we employ the mean squared error (MSE) as the loss functional (denoted by $\mathcal{L}$). Considering losses that additionally model or penalise specific properties of the TV transform may be beneficial to recovering the spectral bands. Therefore, we investigate whether more complex loss functionals are able to recover the spectral bands at a higher image quality. Let $\{b^1(\Theta), \ldots, b^K(\Theta)\}$ be the network output and $\{\hat{b}^1, \ldots, \hat{b}^K\}$ the corresponding ground truth spectral bands for an input image $f$. As the spectral TV decomposition is edge-preserving, we include the normalised Huber loss of the image gradients,

$$\mathcal{L}_\nabla(\Theta) = \frac{1}{K} \sum_{j=1}^{K} \frac{\|\nabla b^j(\Theta) - \nabla \hat{b}^j\|_{Huber}}{\|\nabla \hat{b}^j\|_{Huber}}, \tag{1}$$

to align edges in the bands. Furthermore, using the inverse TV transform introduced in the main paper, we enforce that the sum over all bands is equal to the input image:

$$\mathcal{L}_\sum(\Theta) = \frac{\|\sum_{j=1}^{K} b^j(\Theta) - f\|_2^2}{\|f\|_2^2}. \tag{2}$$

We train four networks with the same parameter and training settings, changing only the loss functionals; we use the MSE loss $\mathcal{L}$ as well as its combinations with the losses introduced in equations (1)

(a) The input image (top) is translated (bottom) to show translational invariance. The spectral bands displayed are cropped to the area of interest, corresponding to a back-translation in the bottom row.

(b) The input image (top) is rotated (bottom) to show rotational invariance.

Figure 5: Visual demonstration of translational and rotational invariances of TVspecNET on a natural image from the BSDS500 dataset [4].

Table 1: Comparison of different loss functionals (cf. equations (1) and (2)) with $\mathcal{L}$ the MSE loss: $\mathcal{L}_1 = \mathcal{L} + \mathcal{L}_\sum, \mathcal{L}_2 = \mathcal{L} + \mathcal{L}_\nabla, \mathcal{L}_3 = \mathcal{L} + \mathcal{L}_\sum + \mathcal{L}_\nabla$. Adding complexity to the loss does not improve performance significantly.

|  | $\mathcal{L}$ | $\mathcal{L}_1$ | $\mathcal{L}_2$ | $\mathcal{L}_3$ |
|---|---|---|---|---|
| SSIM | 0.9600 | **0.9639** | 0.9559 | 0.9619 |
| PSNR | 30.867 | **30.889** | 30.714 | 30.872 |
| sLMSE | **0.8290** | 0.8260 | 0.8224 | 0.8260 |

and (2):

$$\mathcal{L}_1 = \mathcal{L} + \mathcal{L}_\sum, \quad \mathcal{L}_2 = \mathcal{L} + \mathcal{L}_\nabla, \quad \mathcal{L}_3 = \mathcal{L} + \mathcal{L}_\sum + \mathcal{L}_\nabla.$$

The quantitative results for the ablation study are shown in Table 1. We observe no significant improvement in performance compared to the MSE loss. While $\mathcal{L}_1$ has the highest SSIM and PSNR, any variations between losses are very slight. This confirms that the influence of a more complex loss functional on the decomposition retrieval is very limited. The simpler MSE loss is therefore able to recover the spectral TV features and properties without explicitly including this knowledge on TV transform properties at the same high image quality.

## D  Evaluation Metrics

For readers' convenience, we give explicit definitions of the evaluation metrics used throughout the main paper and the Supplemental Material. Let $b$ be an output band of the TVspecNET and $\hat{b}$ the corresponding ground truth band. We evaluate the performance of the TVspecNET on each band individually and average over all bands to obtain an overall image quality value. The three evaluation metrics we use are defined as follows.

**PSNR**    The peak signal-to-noise ration (PSNR) is derived from the MSE between images on a logarithmic scale as follows:

$$\text{PSNR}(b, \hat{b}) = 10 * \log_{10}\left(\frac{\text{MAX}_I^2}{\text{MSE}(b, \hat{b})}\right),$$

where $\text{MAX}_I$ is the maximal possible intensity value for spectral TV bands. Higher PSNR values correspond to a larger signal-to-noise ratio and therefore a better recovery of the ground truth bands.

**SSIM**    The structural similarity index measure (SSIM) [5] describes the similarity between two images based on differences in luminance, contrast and structure. For $\mu_b, \mu_{\hat{b}}$ the mean intensities of images $b, \hat{b}$ and $\sigma_b, \sigma_{\hat{b}}$ their standard deviations, the SSIM is defined as:

$$\text{SSIM}(b, \hat{b}) = \frac{(2\mu_b\mu_{\hat{b}} + c_1)(2\sigma_{b\hat{b}} + c_2)}{(\mu_b^2 + \mu_{\hat{b}}^2 + c_1)(\sigma_b^2 + \sigma_{\hat{b}}^2 + c_2)},$$

where $\sigma_{b\hat{b}}$ denotes the covariance of $b$ and $\hat{b}$, and $c_1, c_2$ are constants that avoid a blow-up when the denominator is small. The SSIM value increases with larger similarity between images and a takes value of 1 for perfect approximation.

**sLMSE**    The inverted localised mean squared error (sLMSE) [2] has been developed to soften the strict MSE metric. Based on the local MSE (LMSE) on image patches, the sLMSE prevents localised errors from dominating the overall image error. For patches $b_\omega, \hat{b}_\omega$ of size $k \times k$ for some estimated and ground truth images $b$ and $\hat{b}$, the sLMSE is defined as

$$\text{LMSE}(b, \hat{b}) = \sum_\omega \|b_\omega - \hat{b}_\omega\|_2^2, \quad sLMSE(b, \hat{b}) = 1 - \frac{LMSE(b, \hat{b})}{LMSE(0, \hat{b})}.$$

In our case, we choose $k = 16$ with a step size of $8$ pixels. sLMSE is an inverted measure, meaning that two images have high similarity if the sLMSE value is close to 1.

## Footnotes

[1] Documentation can be found at https://odlgroup.github.io/odl/