[Reviews · NeurIPS 2020]

Review 1

Summary and Contributions: The objective here is to propose a neural network for spectral TV decomposition of images. The solution relies on a neural network (using an architecture from image denoising) that learns the mapping from image space to TV decomposition.

Strengths: +The proposed solution demonstrate the capacity of convolutional neural network to learn a complex transformation such as spectral TV decomposition +By using a neural network, the time needed to compute the decomposition is reduced by several orders of magnitude. +Experiments are well designed and investigate several interesting aspects: the capacity of the trained NN to learn the underlying properties of this spectral TV transform and different neural network architecture

Weaknesses: After reading the rebuttal and the other reviews I am changing my rating to 6. It is clear that there is an interest in seeing this paper presented at the conference. I hope the authors will integrate the discussion presented in the rebuttal. Weaknesses: -The main weakness is probably the relatively limited contribution: this is a fully supervised setup, using an existing neural network architecture. In addition to this, although the results are compelling, the impact and interest don’t seem sufficient for NeurIPS. -I think spectral TV decomposition is not well known, and one way the authors could (and could have addressed the issue) is to demonstrate some application scenarios where spectral TV decomposition is the best available tool or represents an important preprocessing step. -Another way to address the issue would be to provide a more accessible introduction and background knowledge on spectral TV decomposition. In its current form, section 2 sets the basic equations but goes too much into unneeded technical details, especially given the proposed neural network solution. It would be much more effective to present the high level ideas and the potential of TV decomposition to motivate the reader in using the proposed solution.

Correctness: The experiments seem to support the claim. I didn’t notice any particular issue.

Clarity: Paper is well written but I think the authors could make the background section more interesting for readers not familiar with spectral TV decomposition

Relation to Prior Work: References seem ok

Reproducibility: Yes

Additional Feedback:


Review 2

Summary and Contributions: The paper presents a method for non-linear spectral decomposition of images by approximating the whole pipeline of spectral Total Variation (TV) decomposition with a deep neural network. The proposed approximation is able to capture key properties of the model based decomposition while offering a significant speedup of up to four orders of magnitude.

Strengths: The paper is well written and easy to read. A succinct description of the spectral TV decomposition is given in Section 2, while the proposed method is also briefly but clearly described in Section 3. Important design choices, as for example the choice of DnCNN as a reference architecture in contrast with other well-known architectures, as for example the U-Net and the FFDnet, are theoretically justified and also supported by the evaluation. The experimental evaluation is quite comprehensive (considering also the supplemental material) and shows that the proposed TVspecNET approximates well the spectral TV decomposition while the properties of the model based decomposition (one-homogeneity, rotation and translation invariance) are also captured well. The proposed method potentially has a direct impact on numerous image processing applications including denoising, image fusion and texture separation.

Weaknesses: The main weakness is that the method falls in the category of methods that seek to approximate some function (in this case the PDE of spectral TV decomposition) using a deep neural network. No specially designed architecture is introduced, however the choice of the reference architecture is well motivated as discussed above. Another more practical issue is that by construction the network can only approximate a decomposition into a fixed number of bands. The choice of using dyadically combined bands helps in the evaluation of the decomposition properties, however it would be also interesting to see how close the decomposition by TVspecNET is with respect to the model based one when a larger number of bands is considered. Have the authors considered results for a finer graded decomposition? Is the approximation equally faithful? Finally, for completeness it would be important to provide some additional details regarding DnCNN which is used as the reference architecture for TVspecNET.

Correctness: The claims and the method seem correct. All relevant choices are well justified and the evaluation is quite comprehensive highlighting the contributions of the proposed method.

Clarity: The paper is clear and easy to read. Both the proposed method and the related background are clearly presented.

Relation to Prior Work: Prior work is sufficiently mentioned and discussed.

Reproducibility: Yes

Additional Feedback: Reference to Table 3 should be added in the last paragraph of Section 4.3 L.215: we we # Comments after the rebuttal: The authors addressed my concerns in their response. I think that this is a valuable submission as it proposes a way to perform the spectral TV decomposition much more efficiently than the classical methods. In my opinion the paper should be accepted and, given that the authors will include the clarifications provided in their rebuttal, the paper will be stronger overall.


Review 3

Summary and Contributions: The total variation decomposition of an image can be used to identify and enhance (resp. suppress) structures at a particular scale. This decomposition is expensive to compute using standard methods. Paper describes a method to learn a network that produces a set of 5 bands (and implicitly a residual) from an image. Network and learning are straightforward, but produce an accurate and useful decomposition very much faster than standard methods.

Strengths: This method produces an approximate (but very good) solution to a standard image representation problem very much faster than the standard methods. The approximate decomposition has the properties one expects from the original representation (one-homogeneity; translational and rotational invariance). Curiously, the loss simply requires the network to reproduce examples of bands, rather than (say) imposing some consistency constraint between bands. There is good evidence that the approximate decomposition generalizes, in the sense that the bands produced for out of training images "make sense" as spectral decompositions of the original image. There is overwhelming evidence that the approximate decomposition is very fast.

Weaknesses: The main concern is that, from the point of view of learning, the paper is relatively straightforward. I discount this concern, because the paper produces a representation known to be of interest very much faster than standard methods.

Correctness: Yes

Clarity: Yes

Relation to Prior Work: Yes

Reproducibility: Yes

Additional Feedback: I don't agree with authors (ll274-5) that the main relevance of this paper is to neural network solutions of PDEs. To my mind, the key strength is producing a known useful representation accurately and very much faster than existing methods. ---- added post discussion process ----- I stand by review above, after discussion with other referees.


Review 4

Summary and Contributions: This paper proposes a neural network called TVspecNET that can reproduce the spectral TV decomposition of images while significantly (by more than three orders of magnitude) reducing the computation time to obtain the decomposition once the network is trained.

Strengths: The theory explanation and experiment results reflect the soundness of the claims. The authors propose a deep learning approach to approximate the non-linear spectral decomposition of images as the first ones and significantly speed up the computation of the classical, model driven approach that is based on solving a gradient flow. The method proposed is relevant to the NeurIPS community.

Weaknesses: The description about proposed method (Chapter 3) is a little insufficient, which may make readers difficult to understand the central idea of this paper.

Correctness: The experiments is correct to support their claims and method.

Clarity: Proofreading is still needed, for example, double "we" appear in the line 215.

Relation to Prior Work: In prior work, the decomposition of an image into its TV-spectral bands gives qualitatively highly desirable results its computational realisation is cumbersome as it amounts to the solution of a series of non-smooth optimisation problems. And this paper proposes a neural network approach for obtaining a spectral image decomposition.

Reproducibility: Yes

Additional Feedback:

[Author Response · NeurIPS 2020]

We thank all reviewers for careful reading of our paper and their positive, encouraging, constructive, and in-depth feedback. In the following, we will address the concerns raised in the reviews. Any clarifications given in this author response will be included in the revised version.

**Reviewer #1.**   *Contribution and impact.* Our main contribution is the approximation of a highly non-trivial function by means of deep learning where model-driven approaches have been cumbersome and computationally complex. In that, we decrease the computational requirements substantially. While we do use a well known architecture as the backbone to our approach, we take a step forward in demonstrating intrinsic properties of the non-linear decomposition (e.g. one-homogeneity) are retained and our approach generalises very well to eigenfunctions of the TV subdifferential (very different from the training set). The spectral TV decomposition can be seen as a non-linear analogue to the Fourier transform, and we aim to in some sense obtain an analogue to Fast Fourier Transform (FFT). The availability of a fast computational method is one of the reasons for the success of the Fourier transform in signal and image processing. Fast methods for non-linear spectral decomposition have the potential to become an equally important tool for image and data analysis. *Applications for spectral TV decomposition.* Non-linear spectral decomposition has many successful applications in imaging including denoising [35], texture extraction and separation [6, 23], fusion [4, 46, 20] or segmentation [43] to mention a few. Especially in image fusion applications, spectral TV decomposition is able to overcome challenges in relation to edge, and detail preservation where other methods fail [46]. The use of spectral TV in large-scale applications is hampered by its computational complexity, which is the issue we address in this paper. The advantage and strength of using spectral TV in applications specifically in comparison to, e.g., fully learned approaches is the solid theoretical background. The spectral TV transform is based on the L1 norm and hence is robust, it is convex, edge preserving and does not require parameter tuning. It is mathematically very well understood giving a fundamental non-linear representation, and therefore its use leads to more sound and explainable methods. *Introduction and background.* The reviewer is concerned that the introduction and background are not accessible enough in light of spectral TV decomposition being relatively unknown to the larger community. We will address this issue by adding more high level explanations along the lines of the above. However, we believe that the mathematical details given are necessary to not only understand the complexity of the original problem, but also to introduce the properties known to hold true for spectral TV (such as one-homogeneity) that are later shown to be retained by TVspecNET.

**Reviewer #3.**   *Approximating function with deep neural network.* While we do approximate a function by a neural network, this function is very non-trivial: it is the second derivative of a solution to a highly non-linear and non-smooth PDE, producing a non-linear decompositon of an image. Our main contribution is to use a neural network to reproduce this non-trivial decomposition while retaining intrinsic properties and as a result decrease the computation time by several orders of magnitude. *Fixed number of bands.* As suggested by the reviewer, we trained the same network for a finer graded decomposition with more bands, namely 25 bands, and the results are similarly convincing and faithful (SSIM: 0.958, PSNR: 30.228). However, it is not necessary to retrain the network for a larger number of bands to obtain a finer graded decomposition of certain scales: due to the one-homogeneity one can shift bands in an image by multiplying it by a constant factor (factor <1 will shift larger structures to smaller bands, a factor >1 will do the reverse). Hence, we can represent the structures of interest at a finer scale through shifting without having to retrain the network. *Additional information on DnCNN.* To ensure clarity and completeness, we will elaborate on the network architecture DnCNN that was used as the basis for TVspecNET. *Other issues.* We thank the reviewer for pointing out the missing reference to Table 3 in Section 4.3 and the typo in line 215. We will rectify both issues for the revised version.

**Reviewer #4.**   *Clarification of our contribution.* We thank the reviewer for the remarks on the relevance of our submission. We would like to clarify that we are solving a challenging and non-trivial problem. That is, we approximate the otherwise computationally expensive spectral TV decomposition as it involves solving a highly non-linear and non-smooth PDE, and achieve a much lower computational cost. *Main relevance.* The reviewer pointed out a statement in the broader impact section that upon reflection we see can be easily misunderstood. Instead of claiming the main relevance to be the approximation of the PDE, we intended to highlight that we compute a decomposition that can already be computed with classical methods (and hence the societal impact should be of no concern), but much faster. We will clarify this in the revised version.

**Reviewer #5.**   *Proposed method.* We will take a number of steps to improve the description of our proposed methods to ensure a clear understanding of our central idea. The derivation of the spectral TV decomposition of an image is challenging and computationally expensive as it amounts to solving a highly non-linear and non-smooth PDE and taking the second temporal derivative. We use a neural network to approximate the decomposition at a considerably reduced cost and further show the properties of the analytic TV transform also hold true for TVspecNET. We will add more details about our basis architecture (DnCNN) and extend the high level description by an analogy between the Fourier Transform and spectral TV, and FFT and TVspecNET. *Other issues.* We thank the reviewer for pointing out the typo, we will do a full proof read of the revised version.

[Meta-Review · NeurIPS 2020]

While the proposed method is simple, it addresses a relevant problem and the method performs well and is computationally efficient. Design choices are theoretically justified and the results of extensive experimental evaluation support the claims. After the discussion phase the reviewers agree on acceptance.